# Dynamic Simulation of System Performance Change by PID Automatic Control of Ocean Thermal Energy Conversion

**Lim Seungtaek [1]**, **Lee Hoseang [1]** and **Kim Hyeonju [2,*]**

[1] Seawater Utilization Plant Research Center, Korea Research Institute of Ships & Ocean Engineering, Daejeon 34103, Korea; limst@kriso.re.kr (L.S.); hslee@kriso.re.kr (L.H.)

[2] Offshore Plant Research Department, Korea Research Institute of Ships & Ocean Engineering, Daejeon 34103, Korea

* Correspondence: hyeonju@kriso.re.kr; Tel.: +82-42-866-3700

**Abstract:** Near infinite seawater thermal energy, which is considered as an alternative to energy shortage, is expected to be available to 98 countries around the world. Currently, a demonstration plant is being built using closed MW class ocean thermal energy conversion (OTEC). In order to stabilize the operation of the OTEC, automation through a PID control is required. To construct the control system, the control logic is constructed, the algorithm is selected, and each control value is derived. In this paper, we established an optimal control system of a closed OTEC, which is to be demonstrated in Kiribati through simulation, to compare the operating characteristics and to build a system that maintains a superheat of 1 °C or more according to seawater temperature changes. The conditions applied to the simulation were the surface seawater temperature of 31 °C and the deep seawater temperature of 5.5 °C, and the changes of turbine output, flow rate, required power, and evaporation pressure of the refrigerant pump were compared as the temperature difference gradually decreased. As a result of comparing the RPM control according to the selected PID control value, it was confirmed that an error rate of 0.01% was shown in the temperature difference condition of 21.5 °C. In addition, the average superheat degree decreased as the temperature difference decreased, and after about 6000 s and a temperature decrease to 24 °C or less, the average superheat degree was maintained while maintaining the superheat degree of 1.7 °C on average.

**Keywords:** deep sea water; surface water; ocean thermal energy conversion; closed cycle; proportional–integral–differential controller

## 1. Introduction

Energy consumption is increasing as the size of industry increases with the Fourth Industrial Revolution, and global industrial energy consumption is projected to increase from 222.3 billion Btu in 2012 to 309.1 billion Btu in 2040. Renewable energy is expected to reach 25.1 billion Btu, an increase of 44.2% from 17.4 billion Btu [1]. In addition, the demand for new and renewable energy is increasing in response to environmental problems in the world, and marine renewable energy is also being considered as an alternative [2].

Ocean thermal energy conversion (OTEC) technology generates electricity using a temperature difference between warm water on the ocean's surface and cold water between 800 and 1000 m deep. Potential energy from the temperature difference between the deep water and the surface waters is approximately 1013 W [3]. Surface water exchanges heat with the working fluid to produce steam that acts to operate the turbine. Cold water is used to condense the steam and drive the turbine to the difference in the steam pressure.

In addition, the potential use of seawater temperature differential energy is widespread globally, as shown in Figure 1, with economic feasibility in at least 98 countries [4]. According to Luis Vega of the University of Hawaii, sea temperatures range from 25 °C to 30 °C on the African and Indian coasts, the tropical western and southeastern coasts of the continental United States, and many Caribbean and Pacific islands. In order to solve the global energy shortage, interest and research on OTEC using seawater heat are being conducted [5].

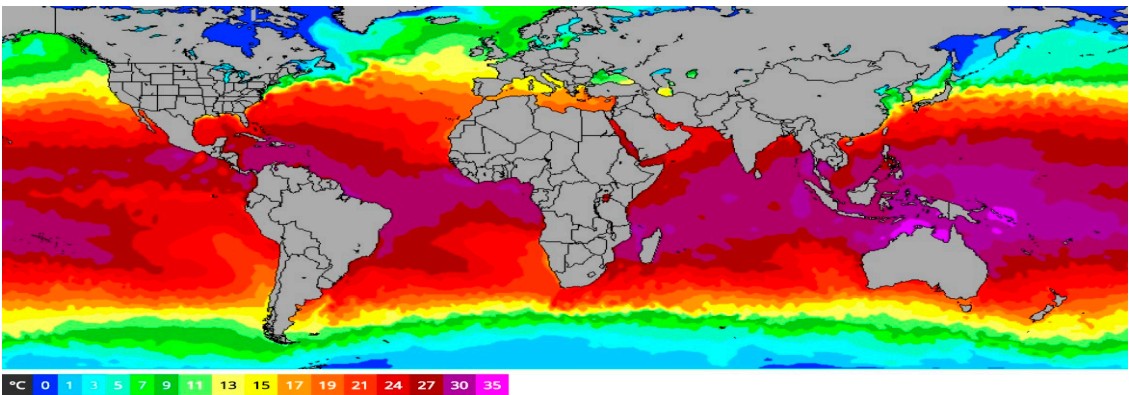

**Figure 1.** Distribution of the world's surface sea temperature (www.seatemperature.org).

The most common power generation method for ocean thermal energy conversion is closed-cycle OTEC power generation using low-temperature refrigerant as a working fluid. Starting with the 50 kW OTEC plant in the US in 1979, a 20 kW pilot plant was built in Korea, the fourth time one in the world, in 2013 [6]. Currently, Korea is producing and demonstrating a 1 MW class demonstration plant for application in Kiribati, a South Pacific island nation. The 1 MW OTEC will include an automatic control system through a proportional–integral–differential (PID) control and an automated operation according to temperature and flow changes. Figure 2 shows the device configuration model for domestic marine demonstration.

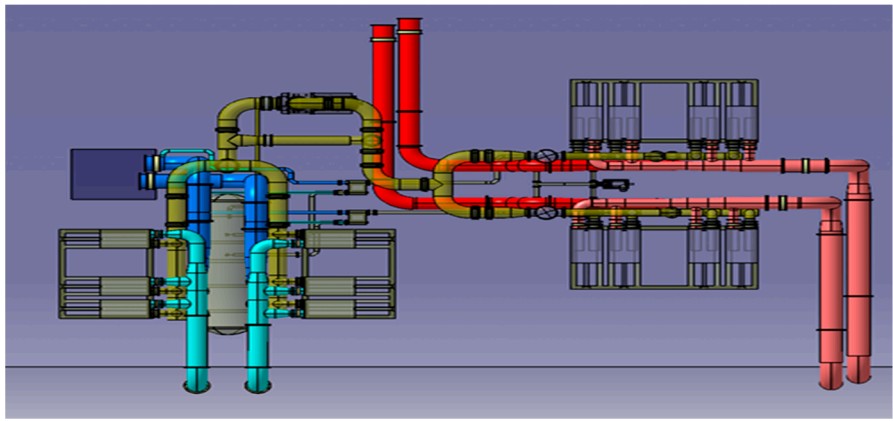

**Figure 2.** Concept design of a 1 MW floating ocean thermal energy conversion (OTEC) in Korea.

On the other hand, studies have been conducted for the automatic control of the demonstration plant in Korea as well as globally. Japan compared the control characteristics of the two-stage OTEC plant with the change of seawater temperature and applied the Proportional-Integral (PI) control to reach the target output. The OTEC with a PI control was reduced to 1/10 of its potential compared to use without a control [7]. In addition, a web-based graphical user interface (GUI) was used to visually monitor the performance change according to the control values [8]. The flow control was established by comparing with the control system of the UEHARA cycle designed in Japan [9,10].

This paper aimed to construct control data by applying the performance data of a turbine, working fluid pump, and heat exchanger to apply a control system for local plant automation prior to a domestic demonstration. In addition, we derived the optimal algorithm and control value for PID control construction.

## 2. Ocean Thermal Energy Conversion Cycle

### 2.1. Closed-Cycle OTEC

Closed-cycle ocean thermal energy conversion (CC-OTEC) generates power by passing working fluids through a turbine and using the temperature difference between the surface and deep water. The working fluid evaporates as the surface water passes through the turbine at high temperature and high pressure and is decompressed and condensed into the low-temperature liquid by the deep water. This cycle is circulated by the working fluid pump. As a heat sink for condensing the refrigerant, deep water of 5 °C or less, which is drawn at a depth of 1000 m or less, is used. Until recently, studies using a closed cycle were actively conducted, and Song [11] studied the application of OTEC plants using seawater intake facilities of Floating Production Storage and Offloading (FPSO) vessels. The phase change refrigerant in the storage is stored in the tank. The heat ($Q_w$) from the evaporation of the OTEC is shown in Equation (1), and the heat ($Q_c$) from the condensation is shown in Equation (2). In addition, the turbine output ($W_t$) of closed power generation is shown in Equation (3), and the net power ($W_{net}$) is shown in Equation (4).

$$Q_w = \dot{m}_{ww}C_p(T_{wwi} - T_{wwo}) \tag{1}$$

$$Q_c = \dot{m}_{cw}C_p(T_{cwi} - T_{cwo}) \tag{2}$$

$$W_t = \dot{m}_r(h_{ti} - h_{to}) = \dot{m}_r\eta_t(h_{ti} - h_{to}) \tag{3}$$

$$W_{net} = W_t - W_{wwp} - W_{cwp} - W_{rp} \tag{4}$$

where $\dot{m}_{ww}$ and $\dot{m}_{cw}$ are the mass flow rate of warm and cold seawater; $C_p$ is the specific heat of seawater; $T_{wwi}$ and $T_{wwo}$ are the temperatures of seawater at the inlet and outlet of evaporator; $T_{cwi}$ and $T_{cwo}$ are the temperatures of seawater at the inlet and outlet of condenser; $m_r$ is the mass flow rate of the refrigerant; $h_{ti}$ and $h_{to}$ are the enthalpy of refrigerant at the inlet and outlet of turbine; $W_{rp}$ is the circulation pump power of the working fluid; and $W_{wwp}$ and $W_{cwp}$ are the pump powers of the warm seawater pump and cold seawater pump, respectively. Closed-cycle OTEC has higher power efficiency and a smaller turbine size than the open type, and it is suitable for constructing large-scale thermal power generation and offshore thermal power generation. The configuration of the closed-cycle OTEC is shown in Figure 3 [12,13].

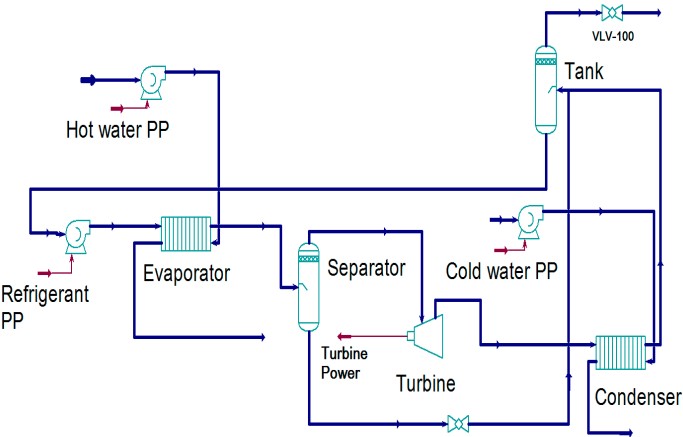

**Figure 3.** A 1 MW OTEC steady-state cycle. (PP: pump).

The working fluid applied to the closed-cycle ocean thermal energy conversion was R32, which is a hydrofluorocarbon (HFC) pure refrigerant, the ozone depleting potential (ODP) value was 0, and the global warming potential (GWP) value was 675, so it had a small environmental destruction factor. In addition, the heat transfer coefficient was approximately 30% higher than R134a and R125, and this allows for the reduction of the heat exchanger volume for the same heat exchange capacity [14,15].

A simulation method using the Aspentech HYSYS (V10) program, which is a process design program, was used to analyze the basic organic Rankine cycle. Helmholtz energy equations, shown in Equation (5), were used. Helmholtz free energy is expressed, in a dimensionless form, as f and expressed by $\int /(RT) = \phi$, the ideal gas part $\phi$, and the residual part $\phi^r$ [16].

$$\frac{\int(\rho, \, \tau)}{RT} = \phi(\delta, \, \tau) = \phi(\delta, \, \tau) + \phi^r(\delta, \, \tau) \tag{5}$$

where $\delta = \rho/\rho_c$, $\tau = T/T_c$, $T_c$ and $\rho_c$ are the temperature and density at the critical point, respectively, and $R$ is the gas constant. The same Helmholtz energy state equation was applied for the design of the closed-cycle OTEC, but the state equation was transformed as shown in Equation (6) and applied according to the characteristics of the R32 working fluid.

$$\phi = \ln \delta + a_0 + a_1 + a_1 \ln \tau + \sum_{i=3}^{6} a_i \ln \left[ 1 - e^{-n_i \tau} \right] \tag{6}$$

where $\alpha$ and $n$ represent coefficients. On the other hand, the state equation of a real gas is shown in Equation (7).

$$\phi^r = \sum_{i=1}^{8} a_i \delta^{d_i} \tau^{t_i} + \sum_{i=9}^{10} a_i \delta^{d_i} \tau^{t_i} \varepsilon^{-\delta^{\varepsilon_i}} \tag{7}$$

where $\varepsilon$ represents the coefficient. The heat balance of the heat exchanger is defined by Equation (8).

$$\text{Balance Error} = (M_{cold}[H_{out} - H_{in}]_{cold} - Q_{leak}) - (M_{hot}[H_{in} - H_{out}]_{hot} - Q_{loss}) \tag{8}$$

## 2.2. Design of the MW OTEC Plant

In order to design the closed-cycle OTEC, the pressure curve and efficiency according to the actual pump and turbine were applied to configure the system. The 1.2 MW class turbine generator manufactured by Jinsol Turbo Machine Co., Ltd. (Korea), has a maximum head of 42.9 m at a flow rate of 120 kg/s using R32 refrigerant as the working fluid and the efficiency is 84.7%. The working fluid pumps of the maker Xylem (USA) vary in efficiency and head as the RPM changes from 345 to 1150. It configured the system to enable RPM control according to changes in external conditions during plant operation. The working fluid pump achieves maximum performance at 1150 RPM with a total head of 46 m and a pump efficiency of 80% at a flow rate of 450 m³/h.

The surface seawater temperature of closed-cycle OTEC was applied to the average temperature of 31 °C in Kiribati. The deep seawater temperature was 5.5 °C, and the actual observation temperature at 1000 m depth in Kiribati was applied. The condensation pressure was 1237 kPa which was sub-cooled to 3.4 °C.

At this time, the flow rate of the working fluid was designed to be 117.7 kg/s, and the surface water and deep water were 1864.6 kg/s and 1503.3 kg/s, respectively. The efficiency of the seawater pump and the efficiency of the turbine were compared by applying the respective performance conditions. The pump efficiency was 79.93%, the pressure increase was 459.6 kPa, and the power consumption was 65.87 kW. The turbine generated 83.06% power generation efficiency, 418.9 kPa pressure change, and 980.5 kW power output.

The head loss from surface and deep water is caused by various factors such as a land riser, heat exchanger, decompression chamber, and fittings. The total head loss of the closed-cycle OTEC was

5.5 m in surface seawater and 8.2 m in deep seawater. The head loss of the OTEC was based on the head design of the closed-cycle OTEC currently being developed by the Korea Research Institute of Ship and Ocean Engineering (KRISO). Equation (9) is a formula for the total head loss which is not considered in this paper, because the static loss and dynamic loss of the closed cycle were too small. The characteristics of the power and head variation with the flow rate and efficiency of the applied turbine and working fluid pump are shown in Figure 4.

$$\sum Total\ headloss = Static loss + Dynamic loss + Pipe\ headloss + Trech\ headloss \\ + Supply\ pipe\ headloss \tag{9}$$

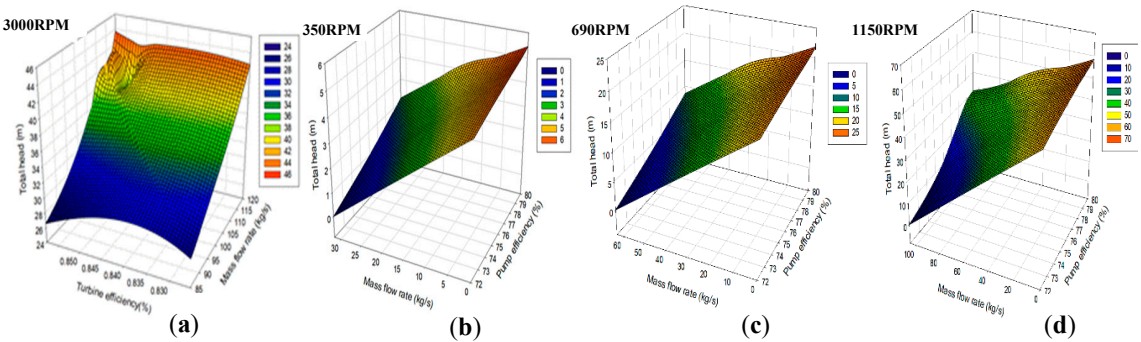

**Figure 4.** The characteristics of the power and head variation with the flow rate and efficiency of the applied turbine and working fluid pump: (**a**) turbine; (**b**) 350 RPM pump; (**c**) 690 RPM pump; and (**d**) 1150 RPM pump.

At this time, the required power of the seawater pump applying the head condition was equal to 158.5 kW and 220.0 kW, respectively.

### 2.3. Dynamic Cycle

2.3.1. Dynamic Cycle Design Theory

Materials Balance

Changes in the physical properties of the flow rate, volume, and density that occur in the system during seawater temperature differential operation are called mass balance, and the mass balance of each component is shown in Equation (10).

$$\frac{d(\rho_{jo}V)}{dt} = F_i\rho_i - F_o\rho_o \tag{10}$$

where $F_i$ is the flow rate of the feed water entering the tank, $F_o$ is the flow rate of the water exiting the tank, $\rho_i$ is the density of the feed water, $\rho_o$ is the density of the discharge water, and $V$ is the fluid volume inside the tank. Through dynamic design, the flow of fluid in and out of the system is caused by convection and diffusion of molecules, contributing to the majority of the flow into and out of the system. Changes in dynamic mass, components, and energy are similar to static state balances which allow the output variables of the system to change over time.

Energy Balance

The change in energy generated or used through OTEC is called the energy balance, and the energy balance of each component is shown in Equation (11).

$$\frac{d[(u + k + \phi)V]}{dt} = F_i \rho_i (u_i + k_i + \phi_i) - F_o \rho_o (u_o + k_o + \phi_o) + Q + Q_r - (w + F_o P_o - F_i P_i) \tag{11}$$

where $u$ is the internal energy, $k$ is the kinetic energy, $\emptyset$ is the potential energy: amount of energy per unit mass, $\omega$ is shaft work energy carried out by the system: energy per hour, $P_2$ is the system pressure, $P_1$ is the pressure of the supply fluid, $Q$ is the thermal energy across the boundary, and $Q_r$ is the reactive thermal energy. The energy flow into and out of the system is by convection or conduction, and the heat added to the system across the boundary is by conduction or radiation. The specifications for the closed-cycle OTEC are shown in Table 1.

**Table 1.** Specifications for the closed-cycle (CC)-OTEC.

| Design Conditions | |
| --- | --- |
| Working fluid | R32, Seawater |
| Warm water temperature | 31 °C |
| Cold water temperature | 5.5 °C |
| Warm water head loss | 5.5 m |
| Cold water head loss | 8.2 m |
| Sea water pump efficiency | 75% |
| Seawater pipe diameter | 1085.6 mm |
| Cold water pipe length | 2870 m |
| Warm water pipe length | 500 m |
| Working fluid pump efficiency | Actual data accepted |
| Turbine efficiency | Actual data accepted |
| Turbine inlet pressure | 1658.7 kPa |
| Turbine outlet pressure | 1239.1 kPa |
| Pump inlet pressure | 1174.6 kPa |
| Warm water flow rate | 1864.6 kg/s |
| Cold water flow rate | 1503.3 kg/s |
| Working fluid flow rate | 117.7 kg/s |
| Turbine inlet temperature | 28.2 °C |
| Condenser heat transfer coefficient | 2084 kJ/°C⁻s |
| Condenser heat transfer area | 761.6 m² |
| Evaporator heat transfer coefficient | 1082 kJ/°C⁻s |
| Evaporator heat transfer area | 897.6 m² |

### 2.3.2. Dynamic Cycle Simulation

Control Range Design

In order to stabilize the system by controlling the flow rate of the working fluid pump, it is necessary to compare the performance change according to the seawater temperature change in advance, and the results are obtained by applying the temperature change of the surface water and the deep water. Kiribati's surface seawater temperature fluctuates annually but falls below 28 °C in winter, while the temperature of deep water remains constant at 5 °C within a 1 °C change annually.

However, the temperature change of the surface water was applied from 26 °C to 31 °C in consideration of the temperature rise due to the fact of extreme weather and leakage from the pipes, and the deep seawater was compared from 5.5 °C to 9.5 °C.

At 1150, the maximum RPM of the working fluid pump, the temperature difference between the surface and deep seawater dropped below 24 °C, and 40 RPM decreased with a temperature change of 1 °C. The superheat was designed to maintain more than 1 °C for the system stabilization, and the

net power output decreased as the temperature gradually decreased, so the system was designed to stop minimum power generation below 50 kW. In addition, the pump RPM and temperature difference were designed to stop below 950 RPM and 19 °C. The configuration of the control flow chart is shown in Figure 5a. The change in performance according to the temperature change of the surface water and deep water is shown in Tables 2–6, and it shows the change in RPM and evaporation pressure, condensation pressure, evaporation temperature and condensation temperature, saturation temperature, superheat and sub-cooling at evaporation and condensation pressure.

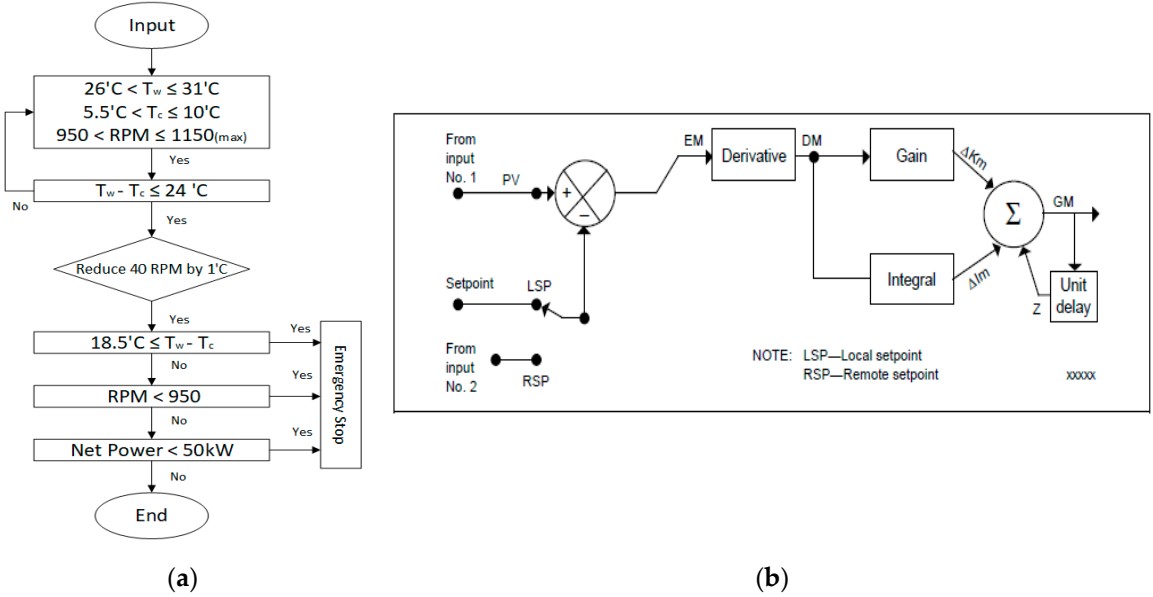

(**a**)                                                                (**b**)

**Figure 5.** Flow chart of dynamic OTEC for PID control and implementation of the PID-A equation: (**a**) flow chart; (**b**) PID-A equation from Honeywell.

**Table 2.** Operating fluid pump RPM control range according to the surface water temperature (cold water temperature: 5.5 °C).

| Warm Water Temperature (°C) | RPM | Evaporating Pressure (kPa) | Condensing Pressure (kPa) | Evaporating Temperature (°C) | Saturating Temperature in Evaporator (°C) | Super Heat (°C) | Condensing Temperature (°C) | Saturating Temperature in Condenser (°C) |
|---|---|---|---|---|---|---|---|---|
| 31 | 1150 | 1687 | 1237 | 28.2 | 24.9 | 3.3 | 10.47 | 13.9 |
| 30.5 | 1150 | 1685 | 1237 | 27.7 | 24.9 | 2.8 | 10.5 | 13.9 |
| 30 | 1150 | 1684 | 1236 | 27.05 | 24.9 | 2.15 | 10.5 | 13.8 |
| 29.5 | 1130 | 1665 | 1231 | 26.75 | 24.5 | 2.25 | 10.3 | 13.7 |
| 29 | 1110 | 1650 | 1225 | 26.3 | 24.1 | 2.2 | 10.1 | 13.5 |
| 28.5 | 1090 | 1630 | 1223 | 26 | 23.7 | 2.3 | 9.8 | 13.5 |
| 28 | 1070 | 1610 | 1220 | 25.6 | 23.2 | 2.4 | 9.65 | 13.4 |
| 27.5 | 1050 | 1600 | 1217 | 25.1 | 23 | 2.1 | 9.5 | 13.3 |
| 27 | 1030 | 1585 | 1213 | 24.7 | 22.6 | 2.1 | 9.25 | 13.2 |
| 26.5 | 1010 | 1570 | 1211 | 24.3 | 22.3 | 2 | 9.1 | 13.1 |
| 26 | 990 | 1550 | 1210 | 23.8 | 22 | 1.8 | 9 | 13.1 |

**Table 3.** Operating fluid pump RPM control range according to the surface water temperature (cold water temperature: 6.5 °C).

| Warm Water Temperature (°C) | RPM | Evaporating Pressure (kPa) | Condensing Pressure (kPa) | Evaporating Temperature (°C) | Saturating Temperature in Evaporator (°C) | Super Heat (°C) | Condensing Temperature (°C) | Saturating Temperature in Condenser (°C) |
|---|---|---|---|---|---|---|---|---|
| 31 | 1150 | 1717 | 1272 | 28.2 | 25.6 | 2.6 | 11.46 | 14.8 |
| 30.5 | 1150 | 1717 | 1271 | 27.6 | 25.6 | 2 | 11.45 | 14.8 |
| 30 | 1130 | 1698 | 1265 | 27.2 | 25.2 | 2 | 11.24 | 14.6 |
| 29.5 | 1110 | 1680 | 1262 | 26.8 | 24.8 | 2 | 11 | 14.5 |
| 29 | 1090 | 1663 | 1258 | 26.4 | 24.4 | 2 | 10.8 | 14.4 |
| 28.5 | 1070 | 1647 | 1255 | 26 | 24.1 | 1.9 | 10.6 | 14.4 |
| 28 | 1050 | 1630 | 1251 | 25.55 | 23.7 | 1.85 | 10.4 | 14.2 |
| 27.5 | 1030 | 1616 | 1247 | 25.15 | 23.4 | 1.75 | 10.24 | 14.1 |
| 27 | 1010 | 1602 | 1246 | 24.72 | 23 | 1.72 | 10.08 | 14.1 |
| 26.5 | 990 | 1588 | 1244 | 24.3 | 22.7 | 1.6 | 9.93 | 14 |
| 26 | 970 | 1575 | 1243 | 23.9 | 22.4 | 1.5 | 9.74 | 14 |

**Table 4.** Operating fluid pump RPM control range according to the surface water temperature (cold water temperature: 7.5 °C).

| Warm Water Temperature (°C) | RPM | Evaporating Pressure (kPa) | Condensing Pressure (kPa) | Evaporating Temperature (°C) | Saturating Temperature in Evaporator (°C) | Super Heat (°C) | Condensing Temperature (°C) | Saturating Temperature in Condenser (°C) |
|---|---|---|---|---|---|---|---|---|
| 31 | 1130 | 1735 | 1277 | 28.2 | 26 | 2.2 | 12.15 | 15 |
| 30.5 | 1110 | 1705 | 1292 | 27.29 | 25.3 | 1.99 | 11.85 | 15.4 |
| 30 | 1090 | 1685 | 1287 | 26.85 | 24.9 | 1.95 | 11.6 | 15.2 |
| 29.5 | 1070 | 1665 | 1285 | 26.45 | 24.5 | 1.95 | 11.4 | 15.2 |
| 29 | 1050 | 1652 | 1282 | 26 | 24.2 | 1.8 | 11.25 | 15.1 |
| 28.5 | 1030 | 1635 | 1280 | 25.55 | 23.8 | 1.75 | 11.05 | 15.1 |
| 28 | 1010 | 1620 | 1276 | 25.1 | 23.4 | 1.7 | 10.9 | 14.9 |
| 27.5 | 990 | 1608 | 1274 | 24.7 | 23.2 | 1.5 | 10.72 | 14.9 |
| 27 | 970 | 1595 | 1273 | 24.25 | 22.9 | 1.35 | 10.55 | 14.8 |
| 26.5 | 950 | 1582 | 1271 | 23.8 | 22.6 | 1.2 | 10.4 | 14.8 |

**Table 5.** Operating fluid pump RPM control range according to the surface water temperature (cold water temperature: 8.5 °C).

| Warm Water Temperature (°C) | RPM | Evaporating Pressure (kPa) | Condensing Pressure (kPa) | Evaporating Temperature (°C) | Saturating Temperature in Evaporator (°C) | Super Heat (°C) | Condensing Temperature (°C) | Saturating Temperature in Condenser (°C) |
|---|---|---|---|---|---|---|---|---|
| 31 | 1090 | 1730 | 1330 | 28.45 | 25.9 | 2.55 | 12.7 | 16.4 |
| 30.5 | 1070 | 1713 | 1325 | 27.4 | 25.5 | 1.9 | 12.6 | 16.2 |
| 30 | 1050 | 1699 | 1321 | 27 | 25.2 | 1.8 | 12.4 | 16.1 |
| 29.5 | 1030 | 1685 | 1318 | 26.55 | 24.9 | 1.65 | 12.2 | 16.1 |
| 29 | 1010 | 1668 | 1317 | 26.15 | 24.5 | 1.65 | 12.05 | 16 |
| 28.5 | 990 | 1655 | 1315 | 25.7 | 24.2 | 1.5 | 11.9 | 16 |
| 28 | 970 | 1642 | 1312 | 25.25 | 23.9 | 1.35 | 11.7 | 15.9 |
| 27.5 | 950 | 1628 | 1310 | 24.8 | 23.6 | 1.2 | 11.5 | 15.8 |

**Table 6.** Operating fluid pump RPM control range according to the surface water temperature (cold water temperature: 9.5 °C).

| Warm Water Temperature (°C) | RPM | Evaporating Pressure (kPa) | Condensing Pressure (kPa) | Evaporating Temperature (°C) | Saturating Temperature in Evaporator (°C) | Super Heat (°C) | Condensing Temperature (°C) | Saturating Temperature in Condenser (°C) |
|---|---|---|---|---|---|---|---|---|
| 31 | 1050 | 1735 | 1360 | 28.6 | 26 | 2.6 | 13.4 | 17.2 |
| 30.5 | 1030 | 1718 | 1355 | 27.6 | 25.6 | 2 | 13.25 | 17 |
| 30 | 1010 | 1705 | 1355 | 27.2 | 25.3 | 1.9 | 13.05 | 17 |
| 29.5 | 990 | 1690 | 1350 | 26.73 | 25 | 1.73 | 12.85 | 16.9 |
| 29 | 970 | 1677 | 1348 | 26.3 | 24.7 | 1.6 | 12.67 | 16.8 |
| 28.5 | 950 | 1663 | 1345 | 25.9 | 24.4 | 1.5 | 12.45 | 16.8 |

PID Control Design

A PID controller is a representative type of control method most commonly used in practical applications. The PID controller basically has the form of a feedback controller. It measures an output of an object to be controlled and compares it with a reference value or a set point desired to obtain an error. It calculates the control value necessary for control using this error value. The standard PID controller is configured to calculate manipulated variables by adding three terms as shown in the following equation. The representing PID control output value (MV(t)) of the RPM according to the time variation is as shown in Equation (12).

$$\text{MV(t)} = K_p e(t) + k_i \int_0^t e(t)dt + K_d \frac{de}{dt} \tag{12}$$

In the above equation, the control parameters $K_p$, $K_i$, and $K_d$ are called gains and represent the proportional value, the integral value, and the differential value, respectively. The magnitude of each term changes with the change of t, a function of time. The error variable ($e(t)$) is generated through the difference between the set point (SP(t)) which is the RPM according to the temperature difference of seawater and the process variable (PV(t)) according to time variation which is expressed as Equation (13).

$$e(\text{t}) = \text{SP(t)} - \text{PV(t)} \tag{13}$$

To derive the optimum PID control value to be applied to OTEC, the range of the variable was converted to a 0%–100% range and then applied to the solution algorithm. The temperature difference between the surface water and the deep water selected as the variable value varied from a maximum of 24 °C to a minimum of 18.5 °C, where the variable value was classified as a percentage through Equation (14).

$$PV(\%) = 100(\frac{PV - PV_{min}}{PV_{max} - PV_{min}}) \tag{14}$$

An upper limit and a lower limit may be specified for all outputs through the determined variable value, and the output limit does not exceed a predetermined minimum or maximum output value. The RPM control ranges were selected for low and high limits of 0% and 100%, respectively, for the output.

In addition, in order to find the optimal proportional value, integral value, and differential value according to each determined operating range, the algorithm was selected, and the difference between the derived RPM and the derived RPM according to the selected RPM was compared through the simulation of the differential value and the integral value.

First, the range of each gain value was changed from 3 s to 5.4 s to select a control parameter so that the response accuracy with the minimum difference between the control value and the set value was compared and the time required to reach the set value was rapid. After applying the controller tuning selection value for the flow control, the temperature difference was compared with the result value by decreasing 1 °C from 24 °C to 23 °C. In this case, the deep seawater temperature was applied at 6.5 °C, and the algorithm used was Honeywell's PID-A model as shown in Figure 5b [17]. When the seawater temperature changes, the error variable increases, and in order to reduce the error, the sudden change of the output is controlled by differential control, and the error is reduced through proportional and integral control gradually with time.

PID Control Value Selection

The description of RPM accuracy and response speed according to the control value change is shown in Figure 6a,b. The RPM accuracy varies greatly with the change in the integral value, with a difference of up to 19.87 RPM at an integral value of 3 s. On the other hand, according to the change of the differential value, the minimum change was 0.41 RPM at the maximum 1.2 RPM. The difference among the optimum operating condition and 1150 RPM was 2.45 RPM which is the minimum value at 5.4 s and 3.6 s, respectively.

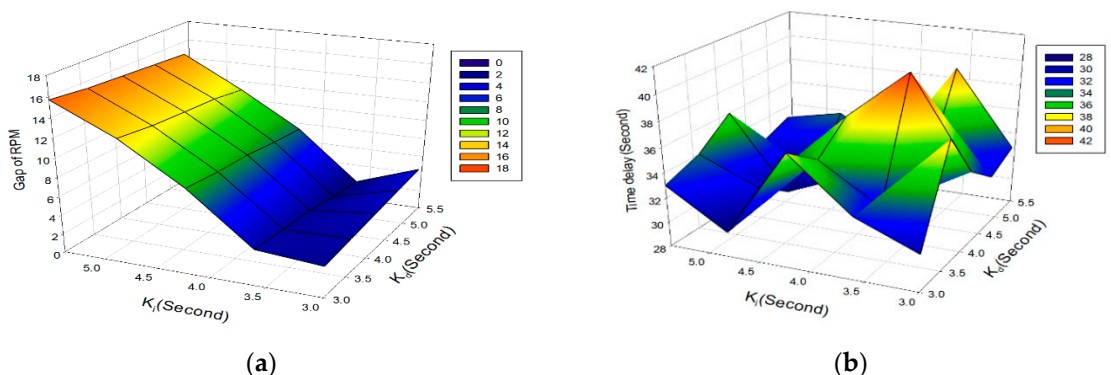

(**a**)                                   (**b**)

**Figure 6.** (**a**) RPM accuracy comparison with the integral and differential and (**b**) comparison of reaction rates with changes in the integral and differential.

On the other hand, the change in response speed, according to the control value, confirmed that the response speed increased rapidly at the specific control value. The minimum response speed was 30 s at the integral and differential values of 4.8 s, and the maximum value was 41 s at the integral value of 3.6 s and the differential value of 4.2 s. The response speed at the control value with the highest

RPM accuracy showed an average of 39 s exceeding 33.6 s, limiting the application the integral value 0.06 min and differential with a response speed of 33 s below the average at the next higher control value. The selection of a control value with a value of 4.8 s was considered.

Dynamic Simulation of OTEC with PID Control

Designing the OTEC with a refrigerant pump control according to the seawater temperature change was done by applying the selected PID value and comparing the difference between the pump RPM value and the design value through the applied control value and by applying the seawater temperature change according to the time change. At this time, it was determined whether the system safety was second by comparing the change in the superheat of the OTEC.

In addition, the power generation amount and the efficiency change were compared by comparing the power generation output and the output change of the working fluid pump. In addition, the control characteristics were verified through comparison with OTEC without automatic control. The operating range for dynamic simulation is shown in Table 7.

**Table 7.** Parameters of the dynamic cycle of the OTEC.

| Parameter | Value | Unit |
|---|---|---|
| Warm water temperature | 30 to 27 | °C |
| Cold water temperature | 5.5 to 6.5 | °C |
| Warm and cold water temperature difference range | 24.5 to 21.5 | °C |
| Pump RPM range | 950~1150 | RPM |
| Pump RPM step per 1 °C | 40 | RPM |
| Working time | 43,000 | Second |
| Reaction time | 36,000 | Second |

## 3. Simulation Result of OTEC

### 3.1. Pump Mass Flow Rate and Superheat Change with Seawater Temperature

### 3.1.1. Without PID Controller

In OTEC operation, which does not include temperature control, the flow rate of the pump cannot be changed automatically as the seawater temperature changes, and the flow rate beyond the design flows into the turbine in the liquid state, even though it passes through the evaporator. The vapor fraction at the outlet of the evaporator decreases rapidly from 1500 s when the temperature difference between the surface water and the deep water reaches 23 °C, and about 10% of the total flow rate flows into the liquid separator. In addition, as the temperature gradually decreases, the evaporator pressure decreases, and the flow rate hunting of the pump occurs. However, from 3500 s, the gaseous refrigerant in the receiver tank flows into the pump which causes an abnormality in the pump. The dynamic change in vapor fraction and flow rate is shown in Figure 7a.

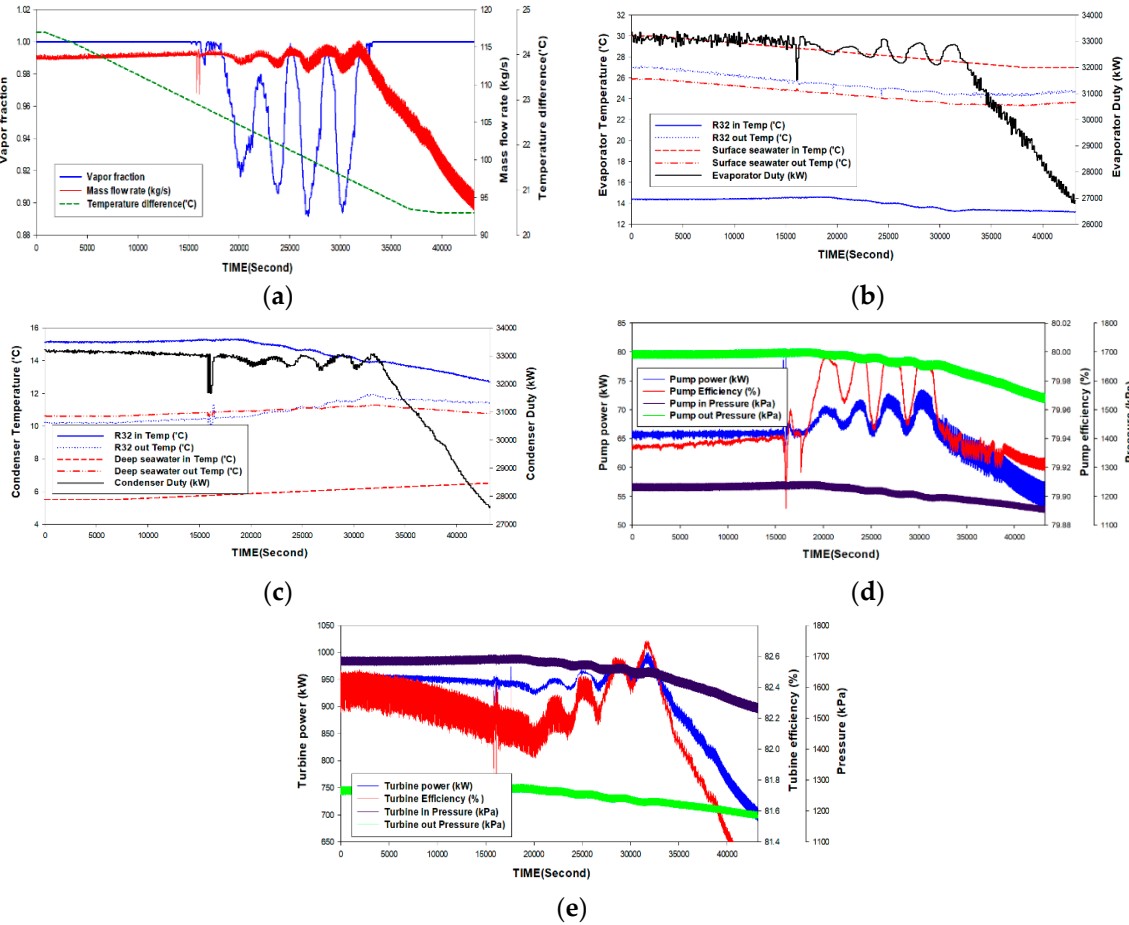

**Figure 7.** Dynamic variations in turbine and heat exchange without PID controller according to the surface and deep water temperature differences: (**a**) vapor fraction and flow rate; (**b**) temperatures and heat capacity of the evaporator; (**c**) temperatures and heat capacity of the condenser; (**d**) power and efficiency of refrigerant pump; (**e**) turbine output and efficiency.

### 3.1.2. With PID Controller

When the seawater temperature was assumed to be 10 h and the surface water was decreased by 3 °C and the deep water was increased by 1 °C, the change of control RPM and the degree of superheating over time were compared. As a result of comparing the RPM control according to the existing PID control value, it was confirmed that the 1010.13 RPM showed the error rate of 0.01% compared to the control value of 1010 RPM in the temperature difference condition of 21.5 °C. At this time, the mass flow rate of refrigerant decreased from the previous 114.2 kg/s to the final 90.2 kg/s. The dynamic change in flow rate and RPM is shown in Figure 8a.

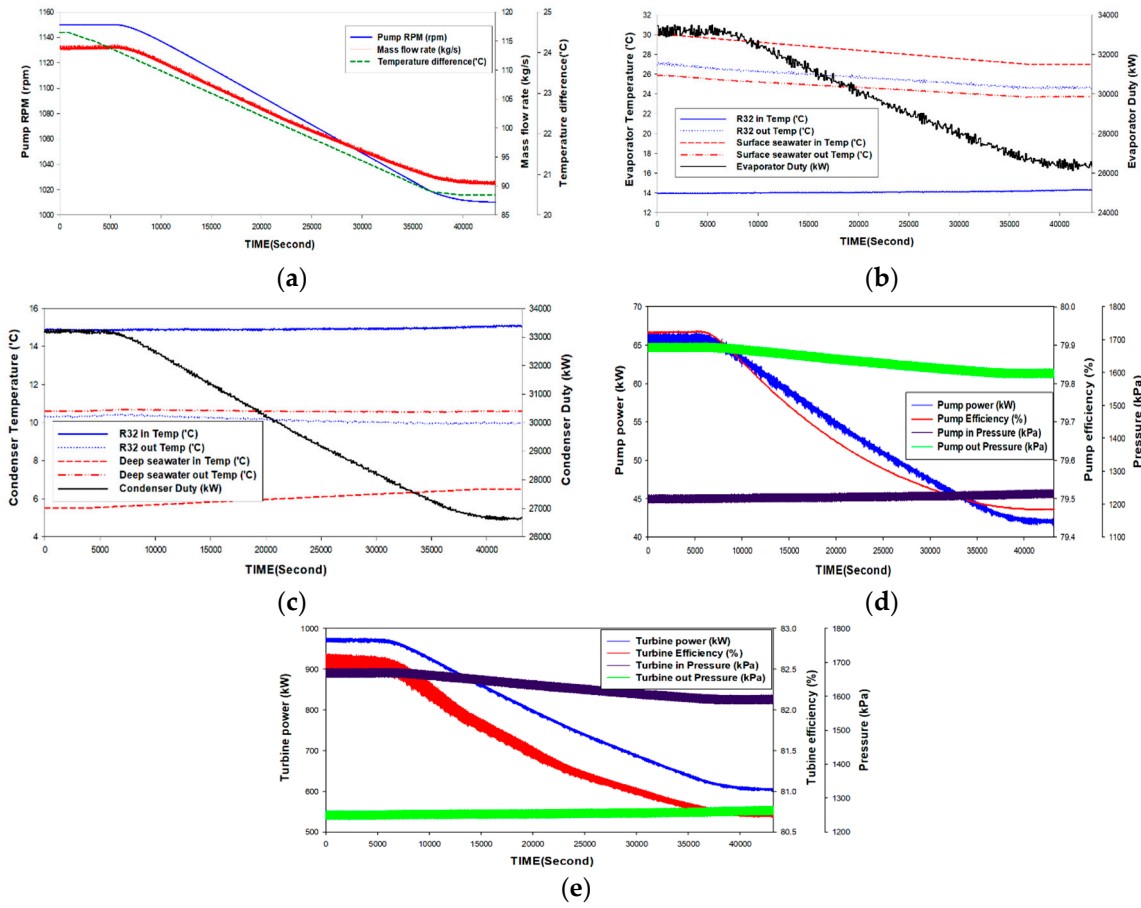

**Figure 8.** Dynamic variations in turbine and heat exchange with PID controller according to the surface and deep water temperature differences: (**a**) refrigerant pump RPM and flow rate; (**b**) temperatures and heat capacity of the evaporator; (**c**) temperatures and heat capacity of the condenser; (**d**) power and efficiency of the refrigerant pump; (**e**) turbine output and efficiency.

### 3.2. Performance Change with Seawater Temperature of OTEC

#### 3.2.1. Without PID Controller

When the seawater temperature was assumed to be 10 h and the surface water decreased by 3 °C and the depth increased by 1 °C, as shown in Figure 7b,c, at a temperature difference of 23 °C, the refrigerant inlet and outlet temperatures in the evaporator decreased and rose repeatedly. In addition, as the gas flowed into the refrigerant pump, the amount of heat was drastically reduced and finally reduced to 27,000 kW or less. On the other hand, in the condenser, the temperature change of the inlet and outlet of the refrigerant decreased from the existing 4.9 °C to the final 1.2 °C.

Figure 7d shows the change in the performance of the refrigerant pump which shows an output of 66 kW and an efficiency of 79.9%, but the power and efficiency were suddenly changed up after the point of 23 °C, and the output increases up to 72 kW, as the seawater temperature difference decreased. However, as the gas entered, the output decreased. The inlet and outlet pressures of the pumps decreased with the changing seawater temperature, and the outlet temperature decreased from the original 1703 kPa to 1524 kPa.

Figure 7e shows the performance change of the turbine with a power output of 959.4 kW and an efficiency of 82.5%. However, four sudden change occurred from 2000 s to 3500 s when the sudden change of the flow occurred, and the output increased up to 1000 kW and then rapidly reduced to 700 kW.

### 3.2.2. With PID Controller

As the surface seawater temperature decreased and the deep water temperature increased the performance change in the seawater temperature difference was compared over time. As the surface seawater temperature decreased and the deep water temperature rose, as shown in Figure 8a,b, the temperature of the refrigerant inlet and outlet in the evaporator decreased, and the temperature difference decreased from the existing 11.9 °C to the final 10.3 °C. Therefore, the calorific value of the evaporator decreased by 22% from 33,504 kW to 26,121 kW. On the other hand, in the condenser, the change in the temperature of the inlet and outlet of the refrigerant was small, from 4.4 °C to the final 4.2 °C.

Figure 8c shows the change in the performance of the refrigerant pump. As the temperature difference between the conventional 66 kW output and the 79.9% efficiency decreased, the output decreased to 41.7 kW, and the efficiency change also decreased to 79.4%. In addition, the pressure changes in the pump inlet and outlet represented 367.1 kPa which is approximately 93 kPa lower than the existing 460 kPa.

Figure 8e shows the performance change of the turbine and a power output of 976 kW and an efficiency of 82.7%. However, as the temperature decreased, the output was reduced to 602 kW, and the efficiency was reduced to 80.7%. The inlet and outlet pressure changes in the turbine were found to decrease from 421.4 kPa to 328.7 kPa.

Figure 9a shows the change in net power and system efficiency according to the sea temperature change of OTEC with automatic control. The OTEC shows a net output of approximately 530 kW at a 30 °C heat source and a 5.5 °C heat sink condition; it decreased to 178 kW with a decreasing temperature difference, and the efficiency decreased from 2.96% to 2.27%.

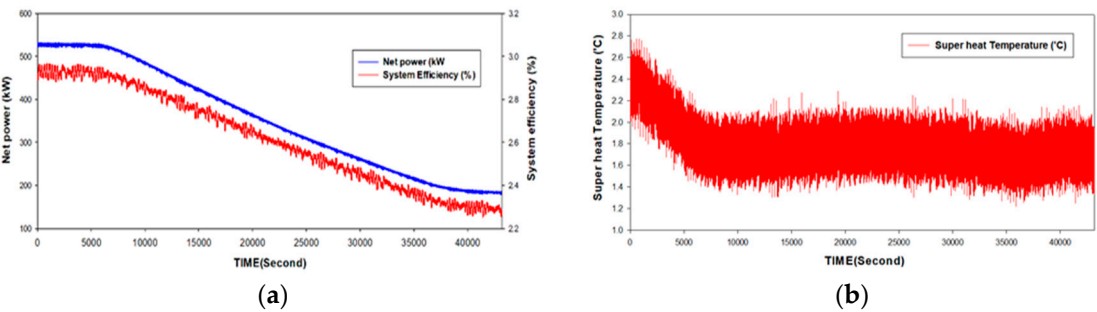

(**a**)                                                                                                                (**b**)

**Figure 9.** Dynamic variations in turbine with PID controller: (**a**) refrigerant superheat of turbine inlet; (**b**) power and system efficiency.

In addition, the change in the superheat, which is the main factor driving the safety of the OTEC, was confirmed as shown in Figure 9b. When the temperature started to change at approximately 720 s after operation, it gradually decreased to a higher position than the control point. Superheat was maintained in the range 1.4~2.2 °C.

## 4. Conclusions

Through this paper, PID control algorithm and control value were selected to build a control system for closed seawater temperature difference generation which is expected to be put into practical use in Kiribati in 2021. In addition, the selected control values were applied to implement the dynamic simulation of the OTEC, and the control accuracy and system stability were examined.

(1)  The RPM control accuracy of the refrigerant pump and the reaction speed were compared by establishing a control system through optimal application of proportional value, integral value, and differential value. The integral value ($K_i$) of 3.6 s and the differential value ($K_d$) of 4.8 s were applied to derive system safety by applying the control system of seawater temperature differential power generation with a 33 s response speed and 2.73 RPM accuracy.

(2) Xylem's working fluid pump applies the efficiency and head change for each flow rate at an RPM of 350, 690 and 1150 and configures the system to control the RPM according to the change in the seawater temperature during plant operation. It was maintained at 1 °C or higher to confirm system stabilization. When the temperature difference between the surface water and the deep water decreased, the average superheat degree (2.3 °C) decreased as time changed. The OTEC system operated while maintaining an average superheat degree of 1.7 °C from approximately 6000 s after falling below 24 °C.

(3) The control algorithm was selected according to the operating characteristics of the pump and turbine to design and control the OTEC plant. By evaluating the responsiveness and accuracy according to the control value and finally evaluating the driving stability through dynamic modeling, it is possible to minimize the control error and unstable operation that can occur in the actual plant.

The control algorithm for the construction of unmanned facilities and control system facilities of the OTEC plant were selected, the control values were derived, and a series of processes from the final simulation to the system verification were established which will be the cornerstone for the future commercialization of OTEC.

**Author Contributions:** Conceptualization, L.S. and K.H.; Methodology, L.S.; Software, L.S.; Validation, L.S. and K.H.; Formal Analysis, L.S.; Data Curation, L.H.; Writing—Original Draft Preparation, L.H.; Writing—Review & Editing, Visualization, L.S.; Supervision, K.H.; Project Administration, Funding Acquisition. All authors have read and agreed to the published version of the manuscript.

**Funding:** This research was funded by a grant from National R&D Project of "Development of 1 MW Ocean Thermal Energy Conversion Plant for Demonstration" funded by the Ministry of Oceans and Fisheries, Korea (PMS4320).

**Conflicts of Interest:** The authors declare no conflict of interest.

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
