# Peer review of "Dynamic Simulation of System Performance Change by PID Automatic Control of Ocean Thermal Energy Conversion"

_jmse, doi:10.3390/jmse8010059_

Round 1

Reviewer 1 Report

The authors simulated close-cycle ocean thermal energy conversion (OCTE) plant automated by PID controllers. The topic is interesting, and the methodology is sound. However, the manuscript does not flow well, and readability must be improved before it can be accepted.

Language needs to be polished. There are several single-sentence, long paragraphs (for example, Line 343-351) that are hard to understand, and a few sentences are incomplete (Line 200, 338). It is suggested to list all the symbols in a separate nomenclature section, as many symbols in the equations are not defined (in Eqs. (1)-(4), Qw, Qc, …). Parameters in Table 1 should be related to symbols in the equations.  It is not necessary to bold symbols. Refer to  format guidelines for more information.   Eq. (13) was repeated after Line 223. The font of legends is too small in Fig. 8. It would be better to split Figure 8 into several groups and make a clear comparison between cases with and without the PID controller. As of now, there is little connection between these plots (Fig. 8(e)-(j)). Line 316: Figure 9(a) was not mentioned.

Author Response

First of all, thanks for the good inquiries and comments. We will respond to the questions and comments pointed out by the reviewer as attached.

Reviewer 2 Report

In this research, automation through PID control is proposed for a closed cycle Ocean Thermal Energy Conversion plant. Dynamic simulations of OTEC were carried our and the control accuracy and system stability were examined. The content of the manuscript is interesting to the readers of Journal of Marine Science and Engineering. However, the following items should be improved.

p.2 line 64: “Reduced to 1/10 level compared to Control free.” is not a sentence.

p.3 line 75: generate -> generates

p.3 eqs(1)-(4): Definitions of many symbols in eqs (1)-(4) are missing.

p.4 line 108: Meaning of the sentence “In order to design the closed-cycle OTEC, detailed design data developed in the 2016 marine offshore plant study was applied, and the pressure curve and efficiency according to the flow rate through the turbine and the working fluid were applied by applying the performance curves of the manufactured turbine and working fluid pump was applied to configure the system.” can not be grasped.

p.4 line 109: was -> were

p.4 line 114: Meaning of the sentence “In addition, Xylum's working fluid pump applied efficiency and head change for each flow rate at RPM 345, 690, and 1150, and configured the system to enable RPM control according to changes in external conditions during plant operation.” can not be grasped.

p.6 line 166: Q Is -> Q is

p.7 line 185: Figure 5 is missing. Moreover, the overall control system can not be explained.

p.7 eqs (12)-(13): Concrete physical values of MV(t), SP(t), PV(t) are not explained.

p.8 Figure 6: The relationship between Figure 6 (a) and (b) can not be understood.

p.8 eq (13): eq(13) is mentioned twice.

p.8 line 226: Figures 8 and 9 -> Figure 7 (a) and (b)

p.8 line 227: 3 seconds -> 3 seconds.

p.9 Figure 7: The units of KI and KD are missing.

p.10 line 258: Figure 10 -> Figure 8(a)

p.10 line 261: the evaporator pressure of the evaporator -> the evaporator pressure

p.13 line 261: In Reference 3, publication year is missing.

p.13 line 277: In Reference 10, title, journal, pages and publication year are missing.

p.14 line 392: Data,, -> Data,

Author Response

(The authors gave the same response as above.)

Round 2

Reviewer 1 Report

The manuscript has been significantly improved and now warrants publication in JMSE.

Author Response

First of all, thanks for the good inquiries and comments. We will upload the final file on the web.

Reviewer 2 Report

Almost all comments are well answered and the manuscript is appropriately revised. Please consider the following small items.

p.7 line 220: Please explain how to determine the setpoint (SP(t)).

p.9 line 279: the -> The

Author Response

First of all, thanks for the good inquiries and comments. We will respond to the questions and comments pointed out by the reviewer as bellow.

Question 1 : p.7 line 220: Please explain how to determine the setpoint (SP(t)).

    1-1. Answer 1 : It was corrected like bellow.

    p.7 line 220: the set point(t) which is the RPM according to the temperature difference of seawater

   2.  Question 2 : p.9 line 279: the -> The

   2-1. Answer 2 : It was corrected.